# DATA-EFFICIENT GRAPH GRAMMAR LEARNING FOR MOLECULAR GENERATION

**Minghao Guo**[1], **Veronika Thost**[2,3], **Beichen Li**[1], **Payel Das**[2,3], **Jie Chen**[2,3], **Wojciech Matusik**[1]

[1]MIT CSAIL, [2]MIT-IBM Watson AI Lab, [3]IBM Research

## ABSTRACT

The problem of molecular generation has received significant attention recently. Existing methods are typically based on deep neural networks and require training on large datasets with tens of thousands of samples. In practice, however, the size of class-specific chemical datasets is usually limited (e.g., dozens of samples) due to labor-intensive experimentation and data collection. This presents a considerable challenge for the deep learning generative models to comprehensively describe the molecular design space. Another major challenge is to generate only physically synthesizable molecules. This is a non-trivial task for neural network-based generative models since the relevant chemical knowledge can only be extracted and generalized from the limited training data. In this work, we propose a data-efficient generative model that can be learned from datasets with orders of magnitude smaller sizes than common benchmarks. At the heart of this method is a learnable graph grammar that generates molecules from a sequence of production rules. Without any human assistance, these production rules are automatically constructed from training data. Furthermore, additional chemical knowledge can be incorporated in the model by further grammar optimization. Our learned graph grammar yields state-of-the-art results on generating high-quality molecules for three monomer datasets that contain only ~20 samples each. Our approach also achieves remarkable performance in a challenging polymer generation task with *only* 117 training samples and is competitive against existing methods using 81k data points. Code is available at `https://github.com/gmh14/data_efficient_grammar`.

## 1 INTRODUCTION

The rise of computational approaches has started to have a significant impact on the discovery of materials and drugs. Recent advances in machine learning, especially deep learning (DL), have driven rapid development on generating novel molecular structures (Maziarka et al., 2020; Xu et al., 2019; Hoffman et al., 2022).Various forms of generative models including generative adversarial networks (De Cao & Kipf, 2018; Maziarka et al., 2020), variational autoencoders (VAEs) (Jin et al., 2018; 2020; Liu et al., 2018; Sattarov et al., 2019), and reinforcement learning (You et al., 2018) have been exploited to represent the complicated molecular design space and generate new molecules. They typically formulate molecular generation as a problem of distribution learning, where the generative model first learns to reproduce the distribution of a training set before generating new molecular structures. Generative models have also managed to integrate certain chemical constraints (e.g., valency restrictions) (Jin et al., 2018; Liu et al., 2018) and shown promising results on the common benchmarks (Irwin et al., 2012; Ramakrishnan et al., 2014). However, DL-based generative models face a serious limitation: they require large amounts of training data to achieve reasonable performance.

In practice, molecule data is not always abundant (Stanley et al., 2021; Altae-Tran et al., 2017; Subramanian et al., 2016); for instance, the focus may be on a specific type of molecules fulfilling certain ingredient requirements. Particularly in the context of polymers, large amounts of training data are not available, and therefore DL models use manually constructed or generated data (St. John et al., 2019; Jin et al., 2020; Ma & Luo, 2020). Real data sets, as used in one of the state-of-the-art papers on polyurethane property prediction, have as little as 20 samples (Menon et al., 2019). In such scenarios, designing a pure DL-based model is challenging.

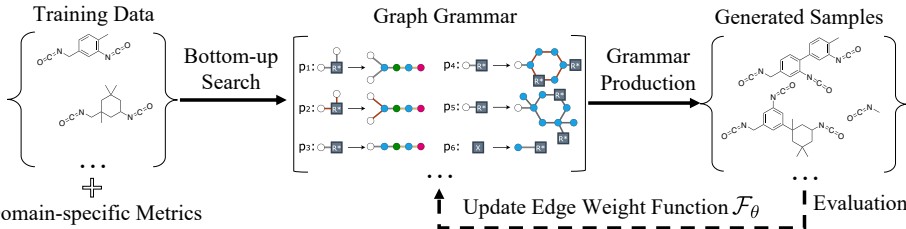

Figure 1: Overview. Given molecules and domain-specific metrics to be optimized, we construct a graph grammar, which can serve as a generative model. The graph grammar construction process automatically learns the grammar rules by optimizing the metrics.

Recent renewed interest in formal grammars (Kajino, 2019; Krenn et al., 2019; Nigam et al., 2021; Guo et al., 2021) provides an alternative to pure DL methods. In formal language theory, a grammar is a set of production rules describing how to generate valid strings according to the language's syntax. A chemical grammar may thus be considered as an interpretable and compact design model that simultaneously serves as a molecular representation and a generative model; even domain-specific constraints can be explicitly incorporated into the rules. Recent examples range from string-based (Krenn et al., 2019; Nigam et al., 2021) to hypergraph-based (Kajino, 2019) and polymer-specific grammars (Guo et al., 2021). Grammar-based generative models do not rely on large training datasets and easily extrapolate to generate molecules outside the distribution of training samples. Yet, they have two major drawbacks. First, current approaches require that chemical grammars are manually designed by human experts (Krenn et al., 2019; Nigam et al., 2021; Dai et al., 2018). This is a tedious process that heavily relies on expertise in chemistry. Moreover, the existing grammars are also very fine-grained (i.e., rules are mostly attaching atoms) in order to cover the syntax of general molecules instead of a specific dataset. Hence, it is not straightforward how to incorporate the bias of given data (e.g., certain molecular substructures) into such grammars. The second drawback is that it remains challenging to integrate chemistry knowledge beyond simple constraints (e.g., valency restrictions) into the grammar. Current grammar construction approaches fail to capture more abstract or complex aspects such as the diversity or synthesizabilty of the generated molecules, which constrains their practicality. Solutions such as Kajino (2019) resort to costly optimization on the level of molecules (or their latent representation) *after* the grammar is built or learned. For a more detailed delimitation of related work, see Sec. 2.

In this paper, we propose a generative model combining complex graph grammar construction with a relatively simple and effective learning technique. In particular, our grammar incorporates substructures of varying sizes (i.e., above atom level) and the construction process directly optimizes various chemical metrics (e.g., distribution statistics and synthesizability) while satisfying specific chemical constraints (e.g., valency restrictions). Moreover, we have the benefits of symbolic knowledge representation: explainability and data efficiency. Our evaluation focuses on polymers, particularly their monomer building blocks. Thus, we curate new, realistic polymer datasets gathered from literature that represent specific classes of monomers. Note that our model works for arbitrary molecules.

**Framework.** Figure 1 outlines our approach. Given molecules and domain-specific metrics to be optimized, we iteratively construct and evaluate a graph grammar as our generative model. We consider the construction as a minimum spanning forest problem and combine it with optimization of the metrics, via a learnable function $\mathcal{F}_\theta$ determining which rules to construct.

**Results.** Our model successfully deals with extreme settings – from a DL perspective – learning meaningful production rules based on only ~10 samples (e.g., of a specific class); this is of significant importance in a practical setting (Stanley et al., 2021; Altae-Tran et al., 2017) but has been ignored so far. In particular, our model is able to generate members of a specific monomer class with a decent success rate. No previous state-of-the-art system achieves similar performance in our experiments. Generally, our approach works on training data that is orders of magnitude smaller than the amount needed by DL-based systems to produce meaningful results. Given ~100 samples, we achieve performance comparable to state-of-the-art systems trained on 81k samples, across a wide range of common and new evaluation metrics. Besides, our grammar optimization method can adjust to any user-defined domain-specific metrics. The learned grammars can capture domain-specific knowledge explicitly, e.g., the characteristic functional groups of a given class of polymers can be extracted from the production rules.

## 2   RELATED WORK

**Generative Models for Molecules.**  Existing models can be categorized based on their molecule representation. It is common to use SMILES strings (Weininger, 1988). Yang et al. (2017), Olive-crona et al. (2017), Chenthamarakshan et al. (2020), Schiff et al. (2021) and Gómez-Bombarelli et al. (2018) use recurrent neural networks to generate the strings in a sequential manner. For instance, Gómez-Bombarelli et al. (2018) propose a VAE based on sequence-to-sequence encoders and decoders. Guimaraes et al. (2017), Sanchez-Lengeling et al. (2017), and Dai et al. (2018) additionally apply optimization objectives to enforce similarity or semantic correctness to generate valid SMILES. Alternatively, molecules can be treated as graphs. Simonovsky & Komodakis (2018), Ma et al. (2018), and De Cao & Kipf (2018) generate the graphs in a single step, directly outputting adjacency matrices and node labels. GraphNVP (Madhawa et al., 2019) uses two steps to generate the node labels based on the adjacency matrix; it exploits a model to encode molecules and uses the same model whose layers are reversed to generate new molecules. You et al. (2018); Li et al. (2018); Samanta et al. (2020); Liu et al. (2018); Liao et al. (2019); Jin et al. (2018; 2020) build molecule graphs iteratively based on either atom nodes or substructures. For example, JT-VAE (Jin et al., 2018) first generates a tree and then expands some of its nodes by attaching substructures based on a vocabulary mined from given molecules; Similarly, HierVAE (Jin et al., 2020) also uses a vocabulary, but improves upon the combinatorial attachment process by directly applying a multi-resolution representation, allowing for generating much larger and diverse molecules. All these methods depend on DL and require large training datasets. Moreover, the resulting models are usually not interpretable. Our approach alleviates this obstacle and can deal with practically common setting of only dozens of available data points.

**Generative Models using Graph Grammars.**  Our approach belongs to the class of models generating new graphs based on the production rules of a graph grammar. These models are non-parametric, interpretable, and can incorporate meaningful graph properties into the rules. One option is to use grammars that are manually designed by human experts (Dai et al., 2018; Nigam et al., 2021). For instance, STONED (Nigam et al., 2021) generates new molecules based on a string-based representation of given molecules by replacing tokens (representing atoms) in accordance with the SELFIES grammar (Krenn et al., 2019). While the simplicity of this approach is attractive, the generated molecules are not necessary reasonable from a chemical perspective (e.g., they might not be synthesizable). This problem can be overcome by mining the grammar from real data. Recently, several automatic techniques that explicitly construct a graph grammar have been proposed in the context of large-scale graphs to facilitate understanding and analysis (Sikdar et al., 2019; Aguinaga et al., 2018; Hibshman et al., 2019). These works are domain-independent and do not allow specialization of the constructed grammar to reflect domain-specific knowledge. Closest to our method is MHG (Kajino, 2019), a framework that constructs a molecular hyperedge-replacement grammar based on Aguinaga et al. (2018). From the given data, MHG learns a fine-grained grammar where the rules iteratively attach single atoms and therefore incorporates hard chemical constraints (e.g., valency restrictions). MHG then applies a VAE conditioned on the grammar to the molecules' tree decompositions in order to also learn soft constraints (e.g., stability). Finally, it uses Bayesian optimization to guide molecule generation. The fine-grained rules in MHG allows for rich diversity. However, our experiments show that these rules make it hard to capture distribution-specific properties, e.g., characteristic substructures of a specific molecular class. Such a drawback limits the practical use of MHG. In contrast, our approach focuses on subgraphs instead of individual atoms only. It directly incorporates domain-specific knowledge into grammar construction and therefore avoids complicated post-processing to achieve high generation performance.

## 3   PRELIMINARIES

**Molecular Hypergraph.**  Molecules can naturally be represented as graphs by taking the atoms as nodes and the bonds as edges. We particularly use a *hypergraph* representation. Given a molecule $M$, the hypergraph $H_M = (V, E_H)$ consists of a set of nodes $V$ and a set of hyperedges $E_H$; a hyperedge can join more than two nodes. Given a regular molecular graph, we construct a hypergraph by including all nodes and adding hyperedges as follows: a hyperedge is added for each bond that joins only two nodes, and for each individual ring (including aromatic ones) that joins all nodes (more than 2) in the ring. Consider Figure 2(a) where we add two hyperedges of the latter kind, one for each ring.

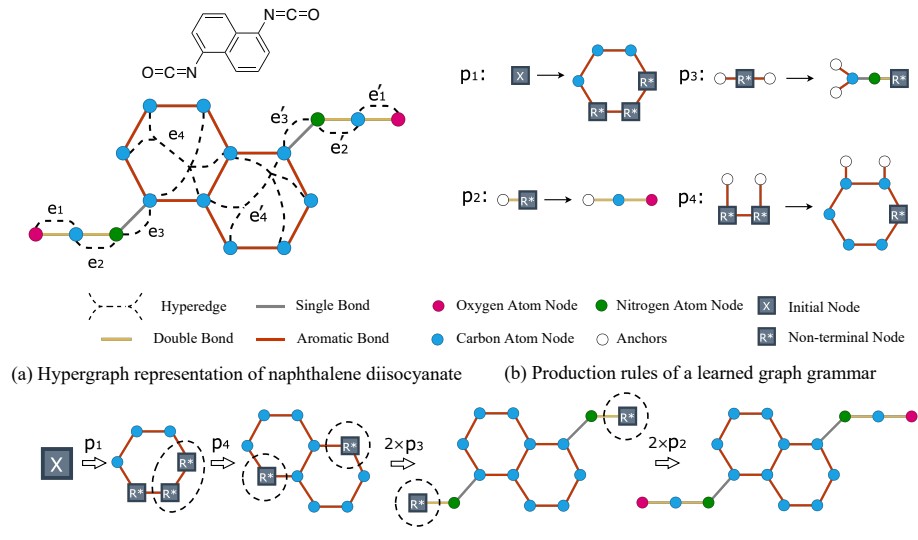

(a) Hypergraph representation of naphthalene diisocyanate    (b) Production rules of a learned graph grammar

(c) Generation process of naphthalene diisocyanate using the graph grammar in (b)

Figure 2: Examples of a molecular hypergraph, one of our possible graph grammars for it, and an application of the grammar for generating new molecules.

**Formal Grammar.** A grammar $G = (\mathcal{N}, \Sigma, \mathcal{P}, \mathcal{X})$ has a finite set $\mathcal{N}$ of non-terminal symbols, an initial symbol $\mathcal{X}$, and a finite set of terminal symbols $\Sigma$. It describes how to build strings from a language's alphabet using a set of production rules $\mathcal{P} = \{p_i | i = 1, ..., k\}$ of form $p_i : LHS \to RHS$, where $LHS$ is short for left-hand side and $RHS$ for right-hand side. Based on such a grammar, a string (of terminal symbols) is generated starting at $\mathcal{X}$, by iteratively selecting rules whose left-hand side matches a non-terminal symbol inside the current string and replacing that with the rule's right-hand side, until the string does not contain any non-terminals.

**Minimum Spanning Forest (MSF).** A minimum spanning tree (MST) is a subset of the edges of a connected, edge-weighted undirected graph that connects all the vertices, without any cycles and with the minimum possible total edge weight. Minimum spanning forest is the union of MSTs for a set of unconnected edge-weighted undirected graphs.

## 4 GRAPH GRAMMAR LEARNING WITH DOMAIN-SPECIFIC OPTIMIZATION

**Graph Grammar.** We focus on a formal grammar over molecule graphs instead of strings, a *graph grammar*. As shown in Figure 2(b), both left and right-hand side of each production rule are graphs. These graphs contain non-terminal nodes (shadowed squares) and terminal nodes (colored circles), representing atoms. The white nodes are *anchor nodes*, which do not change from the left to the right-hand side. Molecule graph generation based on a graph grammar is analogous to the one with string-based grammars described in Sec. 3 (see also Figure 2(c)). To determine if a production rule $p_i$ is applicable at each step, we use subgraph matching (Gentner, 1983) to test whether the current graph contains a subgraph that is isomorphic to the rule's left-hand side. Since the subgraphs are usually small in scale, the matching process is efficient in practice.

**Overall Pipeline.** As shown in Figure 1, the input to our algorithm consists of a set of molecular structures and a set of evaluation metrics (e.g., diversity and synthesizablity). The goal is to learn a graph grammar which can be used for molecule generation. To this end, we first consider each molecule as a hypergraph. The grammar construction is a bottom-up procedure, which iteratively creates production rules by contracting hyperedges (shown in Figure 3). The hyperedges to contract are determined by a parameterized function $\mathcal{F}_\theta$, implemented as a neural network. We simultaneously perform multiple randomized searches to obtain multiple grammars which are evaluated with respect to the input metrics. This approach learns how to create a grammar that samples molecules maximizing the input metrics. Hence, domain-specific knowledge can be incorporated to the grammar-based generative model. The grammar construction and learning process are described in detail in the following Sec. 4.1 and 4.2. The final molecule generation is detailed in Appendix A.

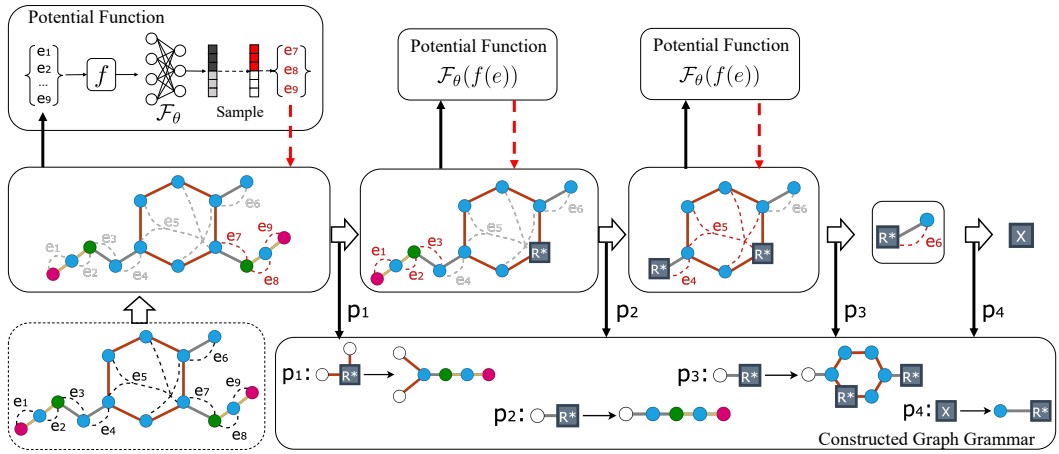

Figure 3: Overview of bottom-up grammar construction. We optimize the iterative, bottom-up grammar construction by learning how to create a grammar that samples molecules fitting input metrics. Specifically we learn which edges to select for contraction in each iteration step using a neural network $\mathcal{F}_\theta$. We perform this construction on all input molecules simultaneously.

## 4.1 BOTTOM-UP GRAMMAR CONSTRUCTION

We describe at a high-level our grammar construction approach (shown in Figure 3). A bottom-up search builds up production rules from the finest-grained level that comprises individual hyperedges in the molecular hypergraph. We construct a grammar by iteratively sampling a set of hyperedges and contracting them into an individual node. The sampling algorithm is described in more detail in Sec. 4.2. For each contraction step, a production rule is constructed and added to the grammar and we obtain a new hypergraph with fewer nodes and edges. We simultaneously perform the hyperedge selection and rule construction for all input molecules until all hyperedges are contracted. Without the loss of generality, we describe the construction process for a single molecule.

At iteration $t$, we consider the current graph $H_{M,t} = (V, E_H)$ and sample $m$ hyperedges $E_t^* = \{e_t^{(i)} \in E_H | i = 1, ..., m\}$. Let $V_t^* = \{v_t^{(i)} \in V | i = 1, ..., n\}$ be the nodes joined by these hyperedges. We then extract all connected components with respect to these hyperedges. Next, we convert each connected component $H_{sub,t}^{(i)} = (V_{sub,t}^{(i)}, E_{sub,t}^{(i)})$ into a production rule. The anchor nodes are those nodes from $V$ that are connected to nodes from $H_{sub,t}^{(i)}$ in the original graph $H_M$ but that do not occur in $H_{sub,t}^{(i)}$ themselves. We also provide notation for the relevant edges:

$$V_{anc,t}^{(i)} = \{v | (s, v) \in E_H, s \in V_{sub,t}^{(i)}, v \notin V_{sub,t}^{(i)}\}, \qquad E_{anc,t}^{(i)} = \{(s, v) | s \in V_{anc,t}^{(i)}, v \in V_{sub,t}^{(i)}\}.$$

Then we construct the production rule $p_i : LHS \to RHS$ with non-terminal node $\mathcal{R}^*$ on the left:

$$
\begin{aligned}
LHS &:= H(V_L, E_L), V_L = \{\mathcal{R}^*\} \cup V_{anc,t}^{(i)}, E_L = \{(\mathcal{R}^*, v) | v \in V_{anc,t}^{(i)}\}, \\
RHS &:= H(V_R, E_R), V_R = V_{sub,t}^{(i)} \cup V_{anc,t}^{(i)}, E_R = E_{anc,t}^{(i)} \cup E_{sub,t}^{(i)}.
\end{aligned}
\tag{1}
$$

When all connected components have been converted into production rules, we update the original hypergraph to $H_{M,t+1}$ by replacing each connected component with the non-terminal node $\mathcal{R}^*$. The above process continues until the hypergraph only consists of one single non-terminal node. For this finally constructed rule, we use the initial node $\mathcal{X}$ instead of $\mathcal{R}^*$ on the left-hand side.

Our proposal has several properties: (1) As a generative model, the grammar can reproduce all input molecules. (2) Since the production rules are constructed from subgraphs of real molecules, valency conditions are naturally obeyed and all generated molecules are thus valid. (3) The generation not only *interpolates* the training data but also *extrapolates* to generate molecular structures outside the distribution of previously seen examples. (4) The constructed grammar roughly follows Chomsky normal form (Chomsky, 1959) and hence is easy to parse and can serve for explanation.

### 4.2 Optimizing Grammar Construction

So far, we have left open a critical question: how to optimize the grammar for the input metrics? Since the grammar construction is *completely* determined by the sequence of hyperedge sets selected, we can convert the optimization of the grammar into the optimization of the hyperedge selection sequence; in particular, note that there are no hyperparameters in addition. Thus, the variable of the optimization problem is the selection sequence with objective to maximize the input metrics.

We formulate the search for a sequence of hyperedge sets as an MSF problem. The bottom-up grammar construction process can be considered as search for a spanning forest for all input graphs. Note that instead of the structure of MSF itself, we focus on the order of hyperedges added to the MSF. The order of hyperedges is determined by an *edge-weight function* $\mathcal{F} : \mathcal{E}_H \to \mathbb{R}$ mapping each hyperedge in every considered molecule hypergraphs to a scalar value. Optimizing the hyperedge selection is equivalent to optimizing the edge weight function. Note that this kind of function is similar to the concept of *potential function* in the field of graphical models (Barber, 2012). Our goal here is to learn a potential function $\mathcal{F}(\cdot)$ that maximizes the given evaluation metrics.

We define our edge weight function as a parameterized function of hyperedge features as $\mathcal{F}(e; \theta) = \mathcal{F}_\theta(f(e))$, where $f(\cdot)$ is a feature extractor function for individual hyperedges $e$, and $\theta$ are the parameters of $\mathcal{F}(\cdot)$ to be optimized. There is no specific restriction on the choice of $f(\cdot)$ so we use a pretrained neural network in our experiments. Our optimization objective is $\max_\theta \sum_i \lambda_i \mathcal{M}_i$, where $\mathcal{M}_i$ is the value of the $i$-th input grammar metric, and $\lambda_i$ is the weight of the $i$-th metric. Since most grammar metrics can only be obtained by evaluating a set of molecules generated by the grammar, it is impossible to get the gradient of $\mathcal{M}_i$ with respect to the parameters $\theta$ via the chain rule. We address this problem by formulating the process of MSF construction as Monte Carlo (MC) sampling. To obtain the gradient of non-differentiable evaluation metrics, we draw inspiration from task-based learning (Donti et al., 2017; Chen et al., 2019) and reinforcement learning, by using REINFORCE (Williams, 1992) to optimize the potential function. Specifically, we define a random variable $X : \Omega \to \{0, 1\}$ on each hyperedge, where $X = 1$ ($X = 0$) means the hyperedge is (not) selected. In each iteration of grammar construction, $X$ follows a Bernoulli distribution, based on the probability $\phi(e; \theta)$ to sample $e$:

$$X \sim \text{Bernoulli}(\phi(e; \theta)), \qquad \phi(e; \theta) = P(X = 1) = \sigma(-\mathcal{F}_\theta(f(e))), \tag{2}$$

where $\sigma(\cdot)$ is the sigmoid function. For $\phi(e; \theta)$, the larger the weight of edge $e$, the lower the probability of $e$ being sampled in the current iteration, which matches the target of MSF. We sample $X$ for each hyperedge and construct grammar production rules iteratively as described in Sec. 4.1. Suppose that the constructed grammar is $G$. Since $G$ is determined by the sampled order of hyperedges, we have $p(G) = p(\mathcal{C}(\mathbf{X})) = p(\mathbf{X}) = \prod_t \prod_j \phi(e_t^{(j)}; \theta)^{X_t^{(j)}} (1 - \phi(e_t^{(j)}; \theta))^{1-X_t^{(j)}}$, where $\mathcal{C}(\cdot)$ is grammar construction process, $e_t^{(j)}$ is the $j$-th hyperedge selected in the $t$-th iteration, and $\mathbf{X}$ is the concatenation of selected hyperedges along all iterations. The optimization objective is:

$$\max_\theta \mathbb{E}_\mathbf{X} \Big[ \sum_i \lambda_i \mathcal{M}_i(\mathbf{X}) \Big]. \tag{3}$$

We estimate the expectation with MC sampling and REINFORCE, approximating the gradient with respect to $\theta$ as:

$$\begin{aligned}
\nabla_\theta \mathbb{E}_\mathbf{X} \Big[ \sum_i \lambda_i \mathcal{M}_i(\mathbf{X}) \Big] &= \int_\mathbf{X} \sum_i \lambda_i \nabla_\theta p(\mathbf{X}) \mathcal{M}_i(\mathbf{X}) \\
&= \mathbb{E}_\mathbf{X} \sum_i \lambda_i \nabla_\theta \log(p(\mathbf{X})) \mathcal{M}_i(\mathbf{X}) \mathrm{d}\mathbf{X} \\
&\approx \frac{1}{N} \sum_{n=1}^N \sum_i \lambda_i \nabla_\theta \log(p(\mathbf{X})) \mathcal{M}_i(\mathbf{X}).
\end{aligned} \tag{4}$$

We then apply gradient ascent in order to maximize the objective. Note that $\mathcal{M}_i(\mathbf{X})$ is normalized to zero mean for each sampling batch to reduce variance in training. The grammar construction, evaluation, and optimization process is repeated until $\theta$ converges or the iteration number exceeds a preset limit. Our approach reduces the complexity of the optimization variable space from being combinatorial (all possible orderings of selected hyperedges) to the dimension of the parameter $\theta$. Such complexity is also much lower than other deep neural network generative models that are directly trained on the molecule dataset.

## 5 EVALUATION

Our evaluation investigates the following five questions:

- How do SOTA models for molecule generation **perform on realistic small monomer datasets**?
- Is our approach effective in **generating specific types of monomers that are synthesizable**?
- How do the models perform **on larger monomer datasets**?
- Can our approach learn to **weigh and optimize different metrics according to user needs**?
- Can our grammar's **explainability support applications**, such as functional group extraction?

### 5.1 EXPERIMENT SETUP

**Data.** We use three small datasets, each representing a specific class of monomers, which we curate manually from the literature: Acrylates, Chain Extenders, and Isocyanates, containing only 32, 11, and 11 samples, respectively (printed in Appendix G). For comparison and for pretraining baselines, we also use a large collection of 81k monomers from St. John et al. (2019) and Jin et al. (2020).[1]

**Evaluation Metrics.** We consider commonly used metrics in the literature (Polykovskiy et al., 2020) and new ones for assessing both individual sample quality and distribution similarity:

- **Validity/Uniqueness/Novelty:** Percentage of chemically valid/unique/novel molecules.
- **Diversity:** Average pairwise molecular distance among generated molecules, where the molecular distance is defined as the Tanimoto distance over Morgan fingerprints (Rogers & Hahn, 2010).
- **Chamfer Distance (Fan et al., 2017):** Distance between two sets of molecules, wherein the usual pairwise Euclidean distance is replaced by the Tanimoto distance.
- **Retro[*] Score (RS):** Success rate of the Retro[*] model (Chen et al., 2020) which was trained to find a retrosynthesis path to build a molecule from a list of commercially available ones[2].
- **Membership**: Percentage of molecules belonging to the training data's monomer class.

We propose the last three as new metrics. We define the Chamfer Distance for molecules to account for "external" diversity between generated data and training data as quantitative indicator of the generative model's extrapolation ability. Ideally, generated molecules should not be too close to any training molecule. Hence, a larger distance is more desired. Note that the commonly used metrics of existing distances only focus on similarity to the training data. Furthermore, we include the Retro[*] Score as a metric estimating sample synthesizability since the commonly used SA Score (Ertl & Schuffenhauer, 2009) does not adequately assess more recent molecules (Polykovskiy et al., 2020). For our small datasets, we check class membership as well. Since a polymer's class is usually determined by its monomer types, it is essential that a generative model can generate class members.

**Baselines.** We compare to various approaches: GraphNVP, JT-VAE, HierVAE, MHG, and STONED; for descriptions see Sec. 2.[3] We call our method DEG, short for Data-Efficient Graph Grammar. Appendix B provides the implementation details.

### 5.2 RESULTS ON SMALL, CLASS-SPECIFIC POLYMER DATA

**Results.** Table 1 shows the results on the Isocyanate data; due to lack of space the other two tables are in Appendix C.1. Observe that GraphNVP has a rather poor performance. The VAEs and existing grammar-based systems perform reasonably well on some metrics, but have low scores on the RS and Membership metrics. In contrast, our method significantly outperforms the other methods in terms of Membership and Retro[*] Score on all three datasets. It also achieves the best or comparable performance on all other metrics.

**Discussion.** Our evaluation shows that learning on monomer datasets, especially when the dataset size is small, is much more challenging than larger datasets as used in the related work (Irwin et al., 2012; Ramakrishnan et al., 2014). GraphNVP shows poor performance since it uses molecular graph adjacency matrix as model input, which is extremely sparse for our relatively large monomer structures. The vocabulary-based JT-VAE and HierVAE perform reasonably well. However, when

---

[1]https://github.com/wengong-jin/hgraph2graph
[2]https://downloads.emolecules.com/
[3]We also tested other generative models (De Cao & Kipf, 2018; Dai et al., 2018) but did not get any meaningful result over our monomer datasets, possibly because these methods are tailored to smaller molecules.

Table 1: Results on Isocyanates, best **bold**, second-best underlined; we omit Novelty since all methods achieved 100%; the few valid molecules generated by GraphNVP did not allow for reasonable evaluation on some metrics (–).

| Method | Valid | Unique | Div. | Chamfer | RS | Memb. |
|---|---|---|---|---|---|---|
| Train data | 100% | 100% | 0.61 | 0.00 | 100% | 100% |
| GraphNVP | 0.16% | – | – | – | 0.00% | 0.00% |
| JT-VAE | **100%** | 5.8% | 0.72 | 0.85 | 5.50% | 66.5% |
| HierVAE | **100%** | 99.6% | 0.83 | 0.76 | 1.85% | 0.05% |
| MHG | **100%** | 75.9% | **0.88** | 0.83 | 2.97% | 12.1% |
| STONED | **100%** | **100%** | 0.85 | 0.86 | 5.63% | 79.8% |
| **DEG** | **100%** | **100%** | 0.86 | **0.87** | **27.2%** | **96.3%** |

the dataset is small, JT-VAE is not able to mine a vocabulary that would allow it to generate many unique molecules. The more diverse vocabulary of HierVAE clearly solves this shortcoming, but the low Membership score shows that it does not capture monomer class specifics. In general, grammar-based methods can better capture class-specific molecule characteristics than DL-based methods and have higher Membership scores. However, they perform poorly on RS. Specifically, MHG has fine-grained rules that simply attaches atoms. This leads to high diversity but the rules hardly capture domain-specific characteristics. The STONED method iteratively replaces atoms based on the SELFIES grammar and only performs interpolation and exploration based on training data, making it hard to embed build-in domain-specific knowledge in the generative model. The overall results show that: 1) Our learned, substructure-based grammar successfully captures class specifics – a critical evaluation criterion which has been ignored so far. 2) Other critical, domain-specific metrics such as RS can successfully be optimized during grammar learning. Our score is $\geq 5\times$ higher than the others. More importantly, the optimization is done in-situ during grammar construction, and hence it avoids post-processing. 3) Our method is the only one that constantly achieves stable performance. Altogether, these results clearly differentiate us from the others.

## 5.3    RESULTS ON LARGE POLYMER DATASET

Our method is developed to model specific classes of complex molecules (e.g., classes of different monomers or polymers) and is expected to deal with most practical scenarios with only a few dozen samples for training. However, as mentioned above, monomer data itself is different from the molecules used in related works. Therefore, we also investigate how our method performs on large monomer datasets comparing with existing methods.[4] Since our approach is relatively complex, but more data-efficient, we apply it to a $0.15\%$ subset. Details are provided in Appendix B.

**Results.** Table 3 in the appendix shows the results. In summary, some SOTA systems such as SMILESVAE and GraphNVP fail to capture any distribution specifics and mostly generate invalid molecules. JT-VAE and grammar-based baselines (MHG, STONED) perform poorly with respect to the former but their sample quality is reasonable. HierVAE performs extremely well on all metrics except Chamfer distance. Our approach can generally compete with the latter (only trained on $0.15\%$ data) and achieves better sample quality, especially Chamfer distance is twice as high.

**Discussion.** Monomer data turns out to be much more challenging compared to the common datasets. Generally, DL-based methods achieve better performance in terms of distribution statistics, while grammar-based models (including ours) have better sample quality. It is reasonable since DL-based methods are all based on distribution learning while grammar-based methods are more focused on modeling chemical rules. Our approach is the only grammar-based system performing well on distribution statistics, which highlights the importance of grammar construction. Fine-grained grammars that either iteratively attaches single atoms (MHG), or only performs input data interpolation (STONED) cannot fit training data more specifically. Our sample quality is among the best. Fur-

---

[4]The data was also used in Jin et al. (2020), but no grammar-based systems were compared.

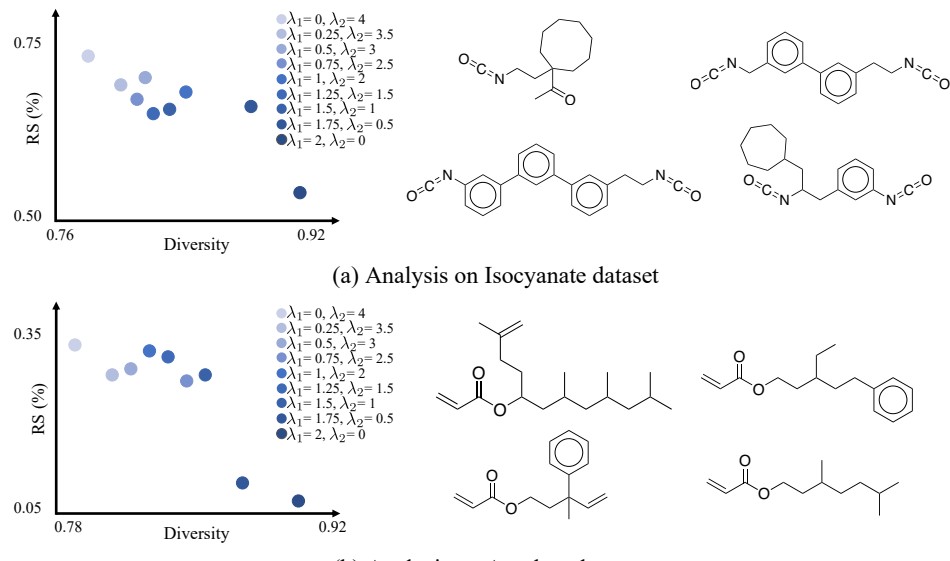

Figure 4: **Left**: Analysis of balance factor $\lambda$. We choose 9 different combinations of $\lambda_i$ for two optimization objectives: Diversity and RS, showing a clear trade-off between the two objectives. **Right**: Examples generated by our learned graph grammar. Our graph grammar can generate novel complex molecular structures that do not exist in the training dataset (e.g., cyclooctane).

thermore, we also show that with more training data (0.3% of the whole dataset), our method can achieve better performance.

## 5.4 ANALYSIS

**Optimizing for Specific Metrics, Balance Factor $\lambda$.** We study the effect of $\lambda$ weighing the importance of metrics according to user needs. We choose 9 different combinations for two optimization objectives: Diversity and RS. $\lambda_1$ ranges from 0 to 2 with 0.25 as interval, while $\lambda_2$ ranges from 4 to 0 with 0.5 as interval. Figure 4 depicts results for Isocyanates and Acrylates. We see that $\lambda$ fulfills its purpose, as the performance of the two objectives can be well controlled. In our study, we use $\lambda_1 = 1, \lambda_2 = 2$ for a balanced trade-off between Diversity and RS.

**Explainability Supports Applications, Functional Group Extraction.** In Appendix F, we show production rules of three graph grammars learned from three small polymer datasets. For each grammar, there is clearly a rule capturing the functional group that characterizes the dataset's corresponding monomer class. For example, $p_3$ is the relevant production rule for Isocyanates. Since functional groups must be present in all monomers of this type, the relevant rule is easily obtained by selecting the rule shared by all inputs.

**Generated Examples.** Figure 4 also shows examples generated using our grammars learned on Isocyanates and Acrylates. Though we have only 32 and 11 training samples respectively, our graph grammar can be used to generate novel and complex molecules. For example, cyclooctane is not contained in the training data but our grammar can generate it by sequentially applying two partial ring formation rules. For more generated examples on the other datasets, see Appendix C.2.

## 6 CONCLUSIONS

We propose a data-efficient generative model combining graph grammar construction with domain-specific optimization. Our grammar incorporates substructures of varying size and the construction directly optimizes various chemical metrics. Extensive experiments on three small size polymer datasets and a large polymer dataset demonstrate the effectiveness of our method. Our system is the only one that is capable of generating monomers in a specific class with a high success rate. It will be useful to incorporate property prediction models with our graph grammar for generating superior molecular candidates for practical use.

ACKNOWLEDGEMENT

This work is supported by the MIT-IBM Watson AI Lab, and its member company, Evonik.

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

## A    DETAILS ON THE MOLECULE GENERATION

Given a learned grammar, we condition the generation strategy based on (non)terminal symbols in the rules: during molecule generation, we exponentially increase the probability of those production rules without non-terminal symbol on the right-hand side based on the iteration number. Formally, the probability to select a certain production rule $r$ at iteration $t$ is $p(r) = Z^{-1} \exp(\alpha t x_r)$, where $x_r$ is a binary value indicating whether rule $r$ contains only terminal symbols on the right-hand side, and $Z$ is a normalization factor. We used $\alpha = 0.5$ in our experiments, since it reduces the generation time sufficiently while maintaining satisfactory diversity.

Note that we initially experimented with uniform random sampling of rules during testing. However, the possibility of generating arbitrarily large molecules (i.e., by choosing production rules with a non-terminal symbol on the right-hand side more likely) sometimes resulted in a never-ending generation process, a problem in practice.

## B    DETAILS ON THE IMPLEMENTATION

**Evaluation Metrics.** For the large polymer dataset, we consider 4 additional evaluation metrics, measuring distribution statistcs of the generated molecules, which are commonly used in DL-based methods. The metrics are:

- **Octanol-Water Partition Coefficient (logP)**: A ratio of a chemical's concentration in the octanol phase to its concentration in the aqueous phase of a two-phase octanol/water system.
- **Synthetic Accessibility Score (SA)** Estimate of how hard or easy it is to synthesize a given molecule based on the molecule's fragments (Ertl & Schuffenhauer, 2009).
- **Quantitative Estimation of Drug-likeness (QED)**: Estimate of how likely a molecule is to be a viable drug candidate, meant to capture aesthetics in medicinal chemistry (Bickerton et al., 2012).
- **Molecular Weight (MW)**: Sum of atomic weights in a molecule. Differences in the histograms for the generated and real data may reveal bias towards generating lighter or heavier molecules.

Note that, while SA Score and QED are not considered useful sample quality heuristics for more recent molecules, they are still useful as distribution statistics (Polykovskiy et al., 2020; Shultz, 2018).

**Baselines.** For STONED, as suggested by the authors, we first generate the chemical space for each sample in the dataset[5].Then we take all samples scoring higher than $0.8$ and randomly sample the remaining ones to obtain our 10k samples. For MHG, we learn a hypergraph grammar on the dataset following the paper's implementation and generate molecules by randomly sampling and deploying production rules. For other baselines we mainly use the settings suggested in the original papers. For the experiments on the small datasets, SMILESVAE, GraphNVP and HierVAE are pretrained on the large dataset and thereafter fine-tuned on specific small dataset; the other baseline implementations do not support pretraining, hence we train them from scratch on the small datasets.

**Our System.** For our approach, we choose a pretrained graph neural network (Hu et al., 2019) as our feature extractor $f(\cdot)$. Note that we do not finetune its parameters during training and it can be replaced by any plug-and-play feature extractors. For the potential function $\mathcal{F}_\theta$, we use a two-layer fully connected network with size $300$ and $128$. For the optimization objectives, we consider two metrics: diversity and RS. For hyperparameters, we set MC sampling size as $5$. We use the Adam optimizer to train the two-layer network with learning rate $0.01$. We trained for 20 epochs.

We train the model and construct grammar on each dataset separately from scratch. For the large polymer dataset, we only use 117 training samples to optimize the grammar. This is motivated by the fact that only 436 different motifs exist in the training dataset as stated in Jin et al. (2020). We can then construct a subset consisting of 436 molecules that can cover these 436 motifs by random sampling. We then sample from this subset to get the training set for our method. For the basic setting of our method, we sample 117 training molecules. We also consider the scenario with more data, where we extend the former set to 239 samples.

After training, we generated 10k samples per dataset for evaluation. For the large polymer dataset, besides the naively generated 10k samples, we further construct another set of generated molecules

---

[5]https://github.com/aspuru-guzik-group/stoned-selfies

Table 2: Results on Acrylates and Chain Extenders (best **bolded**, second-best underlined). The low validity of molecules generated by GraphNVP did not allow for reasonable evaluation on some metrics (–).

| Dataset | Method | Valid | Unique | Div. | Chamfer | RS | Member. |
|---|---|---|---|---|---|---|---|
| | Train data | 100% | 100% | 0.67 | 0.00 | 100% | 100% |
| | GraphNVP | 0.00% | – | – | – | – | – |
| | JT-VAE | **100%** | 0.50% | 0.29 | 0.86 | 4.9% | 48.64% |
| Acrylates | HierVAE | **100%** | 99.7% | 0.83 | 0.89 | 3.04% | 0.82% |
| | MHG | **100%** | 86.8% | **0.89** | 0.84 | 36.8% | 0.93% |
| | STONED | 99.9% | 99.8% | 0.84 | 0.88 | 11.2% | 47.9% |
| | **DEG** | **100%** | **100%** | 0.86 | **0.92** | **43.9%** | **69.6%** |
| | Train data | 100% | 100% | 0.80 | 0.00 | 100% | 100% |
| | GraphNVP | 0.01% | – | – | – | – | – |
| | JT-VAE | **100%** | 2.3% | 0.62 | 0.78 | 2.20% | 79.6% |
| Chain Extenders | HierVAE | **100%** | 99.8% | 0.83 | 0.91 | 2.69% | 43.6% |
| | MHG | **100%** | 87.4% | 0.90 | 0.85 | 50.6% | 41.2% |
| | STONED | **100%** | 99.8% | **0.93** | 0.87 | 6.78% | 61.0% |
| | **DEG** | **100%** | **100%** | **0.93** | **0.94** | **67.5%** | **93.5%** |

Table 3: Results on the large polymer dataset (best **bold**, second-best underlined). The low validity of molecules generated by GraphNVP and SMILESVAE did not allow for reasonable evaluation on some metrics (–). Our method **DEG** was trained on $0.15\%$ and $0.3\%$ of the train data. **DEG** (fitting) is evaluated using the selected 10k samples via the fitting process described in Appendix B.

| Method | Distribution Statistics (↓) | | | | Sample Quality (↑) | | | |
|---|---|---|---|---|---|---|---|---|
| | logP | SA | QED | MW | Valid | Unique | Div. | Chamfer |
| Train data | 0.12 | 0.02 | 0.002 | 2.98 | 100% | 100% | 0.83 | 0.00 |
| SMILESVAE | 9.63 | 2.99 | 0.19 | 751.6 | 0.01% | – | – | – |
| GraphNVP | 2.94 | 0.65 | 0.03 | 435.6 | 0.23% | – | – | – |
| JT-VAE | 2.93 | 0.32 | 0.10 | 210.1 | **100%** | 83.9% | 0.88 | 0.50 |
| HierVAE | **0.50** | **0.08** | **0.02** | **42.45** | **100%** | 99.9% | 0.82 | 0.32 |
| MHG | 9.20 | 1.91 | 0.10 | 380.3 | **100%** | **100%** | **0.91** | 0.56 |
| STONED | 2.43 | 0.81 | 0.07 | 179.9 | 99.9% | **100%** | 0.83 | 0.45 |
| **DEG** (0.15%, fitting) | 1.80 | 0.25 | **0.02** | 69.0 | **100%** | **100%** | 0.82 | 0.60 |
| **DEG** (0.15%) | 5.52 | 0.51 | 0.20 | 334.2 | **100%** | **100%** | 0.86 | 0.62 |
| **DEG** (0.3%, fitting) | 1.93 | 0.23 | **0.02** | 70.2 | **100%** | **100%** | 0.85 | 0.62 |
| **DEG** (0.3%) | 5.64 | 0.41 | 0.19 | 311.4 | **100%** | **100%** | 0.88 | **0.63** |

to have a fair comparison with DL-based methods on distribution statistics. We perform a fitting process selecting 10k data points from 100k generated samples using the same learned grammar in order to get a better performance on distribution statistics. Specifically, we calculate the four evaluation metrics of distribution statistics for both the training dataset and the 100k generated samples;

each data point can have a corresponding 4-dimensional vector. We perform $k$-means clustering of all the vectors of the training dataset with $k = 10,000$. Then we partition the 4-dimensional space into $k$ regions using Voronoi diagram and treat each region 'equal mass'. For each region, we sample one data point from our generated samples that lie in the region. Thus, we construct a set of 10k generated samples and evaluate it the same way as the other methods.

## C    ADDITIONAL RESULTS

### C.1    QUANTITATIVE RESULTS

We report the results on the Acrylate and Chain Extender datasets in Table 2. Since SMILESVAE cannot achieve reasonable validity for all three small datasets, we omit it in the tables. Our method significantly outperforms the others in terms of Retro$^*$ Score and Membership, while achieving best or comparable performance across all other metrics.

The results for the large polymer dataset are reported in Table 3. As common practice (Jin et al., 2020; Polykovskiy et al., 2020), we compute Frechet distance between property distributions of molecules in the generated and test sets for distribution statistics. All but our two new evaluation metrics are computed using MOSES (Polykovskiy et al., 2020). Since the building blocks of polymer data in this dataset are from a special database (St. John et al., 2019), which differs significantly from the emolecule database Retro$^*$ is trained on, we do not report RS as it is less meaningful. Our method achieves remarkable performances across all evaluation metrics. Note that we only use 117 data points while the others are fully trained on the whole dataset.

### C.2    EXAMPLES OF GENERATED RESULTS

(a) Examples generated by the graph grammar learned on large polymer dataset

(b) Examples generated by the graph grammar learned on Chain Extender dataset

Figure 5: Examples of generated results using our learned graph grammar.

## D    SWITCHING THE FEATURE EXTRACTOR

In order to demonstrate the plug-and-play capability of our system's feature extractor, we provide experimental results where we use most embeddings from the deepchem package[6]. As it can be seen in Figure 6, this simple feature extractor yields higher performance than the pretrained GNN at the start of the optimization, but cannot improve further through the whole learning process. In order to obtain a reasonable optimized grammar, an ideal feature extractor should capture both the local and

---

[6]https://deepchem.readthedocs.io/en/latest/api_reference/featurizers.html (we just concatenated atom and bond features)

the global information of the hyperedges in the hypergraph. Hence, the GNN fits better our purpose of grammar construction.

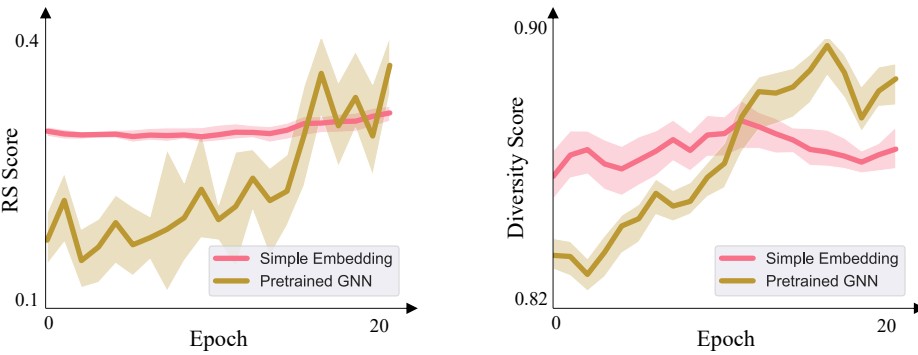

Figure 6: Comparison with simple feature extractor on Isocyanates.

# E   DEMONSTRATION OF STABILITY OF REINFORCE

Since REINFORCE is known to have large variance, we provide results on the stability of our proposed DEG. We run our algorithm with three random seeds on the Isocyanates data. The experimental results are shown in Figure 7. The diversity scores are relatively stable across the three random seeds. The RS scores vary more but all three experiments converge to a similar value around the end of optimization. We also show convergence curves for the other three datasets in Figure 8.

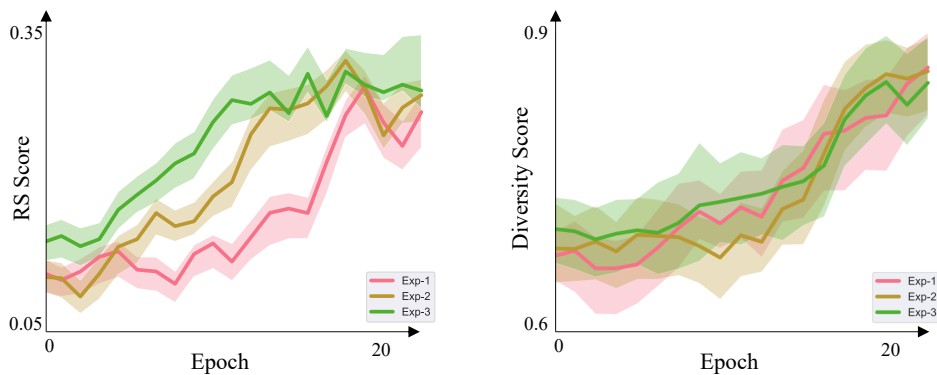

Figure 7: Analysis of stability of proposed DEG.

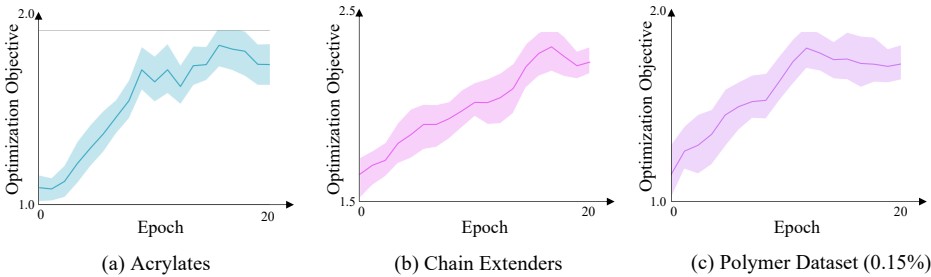

(a) Acrylates        (b) Chain Extenders        (c) Polymer Dataset (0.15%)

Figure 8: Convergence curves on three datasets.

## F  LEARNED GRAPH GRAMMAR

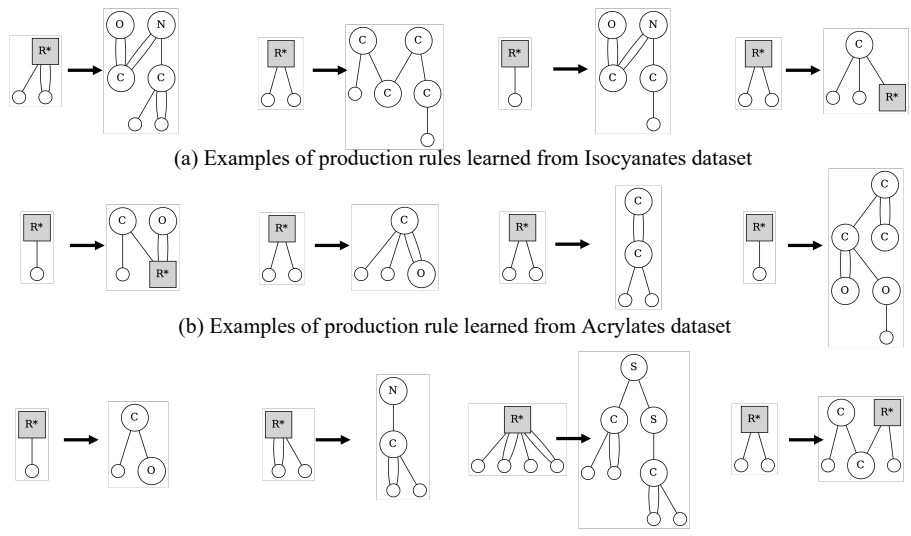

(a) Examples of production rules learned from Isocyanates dataset

(b) Examples of production rule learned from Acrylates dataset

(c) Examples of production rule learned from Chain Extenders dataset

Figure 9: Examples of production rules from our learned graph grammar.

## G  OUR DATASETS

### G.1  ISOCYANATES

MDI  `O=C=NC1=CC=CC(CC2=CC=C(C=C2N=C=O)CC3=CC=C(C=C3)N=C=O)=C1`

MDI  `O=C=NC1=CC(CC2=C(C=C(C=C2)CC3=CC=C(C=C3N=C=O)CC4=CC=C (C=C4)N=C=O)N=C=O)=CC=C1`

MDI  `O=C=NC1=CC=C(C=C1)CC2=CC=C(C=C2N=C=O)CC3=C(C=C(C=C3)CC4=C C=C(C=C4N=C=O)CC5=CC=C(C=C5)N=C=O)N=C=O`

HDI  `O=C=NCCCCCCN=C=O`

HDI  `O=C=NCCCCCCCCCCCCCN=C=O`

HDI  `O=C=NCCCCCCCCCCCCCCCCCCCN=C=O`

HDI  `O=C=NCCCCCCCCCCCCCCCCCCCCCCCCCN=C=O`

IPDI  `CC1(CC(CC(CN=C=O)(C1)C)N=C=O)C`

TDI  `CC1=C(C=C(C=C1)CN=C=O)N=C=O`

HMDI  `O=C=NC1CCC(CC2CCC(CC2)N=C=O)CC1`

LDI  `CCOC(C(N=C=O)CCCCN=C=O)=O`

### G.2  ACRYLATES

Benzyl Acrylate  `C=CC(=O)OCC1=CC=CC=C1`

Phenyl Acrylate  `C=CC(=O)OC1=CC=CC=C1`

Phenyl Methacrylate  `CC(=C)C(=O)OC1=CC=CC=C1`

2-Phenylethyl Acrylate  `C=CC(=O)OCCC1=CC=CC=C1`

n-Octyl Methacrylate  `CCCCCCCCOC(=O)C(=C)C`

Sec-Butyl Acrylate  `CCC(C)OC(=O)C=C`

Benzyl Methacrylate  `CC(=C)C(=O)OCC1=CC=CC=C1`

Pentafluorophenyl acrylate  `C=CC(=O)OC1=C(C(=C(C(=C1F)F)F)F)F`

iso-Butyl methacrylate `CC(C)COC(=O)C(=C)C`

n-Dodecyl methacrylate `CCCCCCCCCCCCOC(=O)C(=C)C`

sec-Butyl methacrylate `CCC(C)OC(=O)C(=C)C`

n-Propyl methacrylate `CCCOC(=O)C(=C)C`

3,3,5-Trimethylcyclohexyl methacrylate `CC1CC(CC(C1)(C)C)OC(=O)C(=C)C`

iso-Decyl acrylate `CC(C)CCCCCCOC(=O)C=C`

n-Propyl acrylate `CCCOC(=O)C=C`

2-Methoxyethyl acrylate `COCCOC(=O)C=C`

2-Phenoxyethyl methacrylate `CC(=C)C(=O)OCCOC1=CC=CC=C1`

n-Hexyl acrylate `CCCCCCOC(=O)C=C`

2-n-Butoxyethyl methacrylate `CCCCOCCOC(=O)C(=C)C`

Methyl Methacrylate `CC(=C)C(=O)OC`

Methyl Acrylate `COC(=O)C=C`

Butyl Arylate `CCCCOC(=O)C=C`

2-Ethoxyethyl methacrylate `CCOCCOC(=O)C(=C)C`

Isobornyl methacrylate `CC(=C)C(=O)OC1CC2CCC1(C2(C)C)C`

2-Ethylhexyl methacrylate `CCCCC(CC)COC(=O)C(=C)C`

Neopentyl glycol propoxylate diacrylate `CC(C)(COCCCOC(=O)C=C)COCCCOC(=O)C=C`

1,6-Hexanediol diacrylate `C=CC(=O)OCCCCCCOC(=O)C=C`

Pentaerythritol triacrylate `C=CC(=O)OCC(CO)(COC(=O)C=C)COC(=O)C=C`

Trimethylolpropane propoxylate triacrylate
`CCC(COCCCOC(=O)C=C)(COCCCOC(=O)C=C)COCCCOC(=O)C=C`

Di(trimethylolpropane) tetraacrylate
`CCC(COCC(CC)(COC(=O)C=C)COC(=O)C=C)(COC(=O)C=C)COC(=O)C=C`

Dipentaerythritol pentaacrylate
`C=CC(=O)OCC(CO)(COCC(COC(=O)C=C)(COC(=O)C=C)COC(=O)C=C)COC(=O)C=C`

Dipentaerythritol hexaacrylate
`C=CC(=O)OCC(COCC(COC(=O)C=C)(COC(=O)C=C)COC(=O)C=C)`
`(COC(=O)C=C)COC(=O)C=C`

## G.3 CHAIN EXTENDERS

EG `OCCO`

1,3-BD `OC(C)CCO`

BD `OCCCCO`

AE-H-AE `OCCNC(=O)NCCCCCCNC(=O)NCCO`

AE-L-AE `OCCN1C(=O)NC(C1(=O))CCCCNC(=O)NCCO`

D-E-D `Oc1ccc(cc1)CCC(=O)OCCOC(=O)CCc1ccc(cc1)O`

LYS `OC(=O)C(N)CCCCN`

L-Orn `OC(=O)C(N)CCCN`

Pip `N1CCNCC1`

AFD `Nc1ccc(cc1)SSc2ccc(cc2)N`

MDA `Nc1ccc(cc1)Cc2ccc(cc2)N`

