# OpenReview forum: "Data-Efficient Graph Grammar Learning for Molecular Generation"
_ICLR.cc/2022/Conference — ICLR 2022 Oral_

### Official Review · Reviewer_LtXp · 2021-10-21

**Correctness:** 4
**Technical Novelty And Significance:** 4
**Empirical Novelty And Significance:** 3
**Recommendation:** 8
**Confidence:** 4

**Main Review:**

Strength:
This method is very data efficient given grammar-based learning.
The hypergraph contraction-based production rule learning process is very novel (at least from my perspective).

Weakness:
It seems like the model can only work on a small set of molecules to generate the rules due to the computational complexity.
The way this model learns the production rule is actually very related to a domain called task-based learning, where people tend to optimize their model towards some non-differentiable evaluation metrics.
The authors should reference some of those works, e.g.
"Task-Based Learning via Task-Oriented Prediction Network with Applications in Finance",
"Task-based End-to-end Model Learning in Stochastic Optimization".

**Summary Of The Paper:**

This paper proposed an interesting grammar learning-based data-efficient molecule generation model.
The key idea of this method is to learn a set of production rules via a hyper-edge selection process that optimizes a set of evaluation metrics.
This method achieves impressive performance on extremely small datasets and achieves competitive performance compared with end-to-end models trained on a large dataset.


**Summary Of The Review:**

Due to the novelty of this data-efficient grammar-based molecule generation method and its impressive performance, I recommend to accept this paper.

---

> ### Author Response · Authors · 2021-11-19
> **We thank the reviewer for their interesting comments! Please see our rebuttal below.**
>
>
> - **Model Works "Only" on Small Datasets** \
> Our goal was to design a molecule generation model for the "few data" scenario, for which there are no solutions yet. It is indeed true that our approach solves this challenge with added complexity - amongst others, by applying symbolic knowledge representation which, besides having several benefits, is known to be more involved. For practical scenarios where the data is too large for our model (~1k samples), we recommend to use other deep learning-based methods.
> \
> \
> Nevertheless, there are several important points on optimization to be considered in practical applications. First, the convergence of learning can be accelerated by switching the naive MC sampling to a more advanced method (e.g., importance sampling [3]). Further, the implementation can be improved. System-level code optimization has been out of our focus so far but we expect considerable acceleration can be achieved by adding parallelization.
> \
> \
> Lastly, we want to stress that, unlike what the wealth of DL-based models suggests, the small data scenario is much more common in practice [4]:
> > even though such big datasets (and access to them) are becoming common place, they do not represent the datasets most materials researchers work with on a day-to-day basis. Within general experimental material research projects, researchers generally produce no more than a hand full of data points (c.q. samples) when optimizing a production method, synthesizing a new material or tuning an existing one for a specific application.
>
>
> - **Related Work on Task-based Learning** \
> Thank you for pointing out this connection. We added some such works in the updated paper.
>
>
> [3] Feller, William. An introduction to probability theory and its applications, vol 2. John Wiley & Sons, 2008. \
> [4] Vanpoucke et al. Small data materials design with machine learning: When the average model knows best  featured, Journal of Applied Physics 128, 054901 (2020).

---

### Official Review · Reviewer_Qrrj · 2021-10-23

**Correctness:** 4
**Technical Novelty And Significance:** 3
**Empirical Novelty And Significance:** 3
**Recommendation:** 8
**Confidence:** 3

**Main Review:**

This is a pretty solid contribution. Although the paper is written clearly, I admit I had trouble trying to understand the method initially, but once I got the big picture, everything lined up nicely. The methodology in itself (generative learning through graph grammars) is not new, and the proposed model presents some similarities to the HMG model of Kajino. However, it draws from previous methods in a clever way and addresses most of their limitations (for example, the production rules involve substructures instead of single atoms). This, in my opinion, is a plus. The main advantage of this model is its data efficiency, which the experiments corroborate nicely. Nonetheless, I have a few points I would like the authors to address before I give complete acceptance.

1) It is unclear to me what is the computational cost of constructing the grammar and training the model since generating the production rules involves graph matching (and although you affirm this isn't an issue). How does your training/grammar inference cost compares to other approaches?

2) "we also show that with more training data (0.3% of the whole dataset), our method can achieve better performance". I might have missed it, but where is this shown?

3) Why are some results in the tables underlined? You should explain it in the captions.

4) This is more of a request for clarification. How is the final set of grammar rules chosen? If I understood correctly, the bottom-up construction process generates a lot of near-duplicates, as well as potentially useless production rules, especially during the initial iterations. Is there any pruning process after the optimization has been performed, or just everything is kept?

5) How many rules does the grammar construction process produce in the datasets you evaluated?

**Summary Of The Paper:**

The paper presents a data-efficient graph grammar-based approach to molecular generation (with a focus on polymer generation). The model requires that the molecule be represented as a hyper-graph. The productions of the grammar are basically rules for contracting connected nodes into a single node (a non-terminal symbol), which are later expanded during generation. Each rule is essentially derived from a random sample of the hyper-graph edges: the grammar is produced by iteratively contracting the nodes incident to the edges into non-terminal nodes until the hyper-graph consists of a single non-terminal node. This procedure is applied simultaneously to all the graphs of the training set. Once a grammar is obtained, it is optimized by maximizing some desired metrics (such as diversity). The optimization is non-differentiable since it requires generated graphs to evaluate the objective function, and thus the authors resort to approximate the gradient's expectation using MC sampling and the REINFORCE score function estimator.
The experiments aim at showing that the proposed model is data-efficient (achieving strong performances using a small fraction of the molecules used by competitors), capable of extrapolation (since it can produce molecules outside of the training set), and explainable (since it is shown to identify functional groups which characterize the family of generated molecules).



**Summary Of The Review:**

I am decidedly leaning towards acceptance. In my opinion, this is a strong contribution that deserves to appear at ICLR. I will most likely raise my score to an 8 after the authors will answer my questions and modify the text accordingly where needed.

---

> ### Author Response · Authors · 2021-11-19
> **We thank the reviewer for their interesting comments! Please see our rebuttal below.**
>
>
> - **(Q1) Computational Cost**
> \
> Although we did not optimize our implementation, which could be substantially accelerated, we note that for the small data scenario, the current runtimes are quite manageable: for the 117-sample training set, our model takes 1 hour per training epoch and we train it for 20 epochs (the 200 in the appendix was an unfortunate typo).
> \
> \
> Since there are no existing solutions for the scenario (small training data) we address, we can only compare to the models addressing the large data scenario, which obviously requires considerable training efforts. Amongst those approaches, our approach is closest to the vocabulary (Jin et al., 2018 & 2020) and grammar-based models (Dai et al., 2018; Kajino, 2019; Nigam et al., 2021). The vocabulary-based models extract the substructure vocabulary in a preprocessing step. At runtime, Jin et al. (2018) apply enumeration to assemble substructures, whose time complexity is exponential to the substructure size and heavily relies on assumptions of the nature of the substructures. HierVAE (Jin et al., 2020) also took several days to (pre)train on the large dataset. Dai et al. (2018) and Nigam et al. (2021) apply a manually constructed grammar, which is not at all comparable to our grammar that captures the specifics of the dataset at hand. Theoretically, the grammar construction of MHG (Kajino, 2019) is not polynomial anymore since it also applies a graph isomorphism test. Yet, in practice it runs rather fast since the system simply decomposes everything to the finest level; as a result, the grammar does however not capture the critical substructures. Overall, we note that this comparison is lacking since the existing works are of very different nature (see also the reply to reviewer LtXp).
>
>
> - **(Q4) Clarification about Algorithm**
> \
> Please see "Details about Molecule Generation using the Learned Grammar" and "Pruning" in the reply to reviewer oHTg, where we detail that the constructed grammar is *completely* determined by the optimization process (described in Sec. 4.2). We only remove duplicate rules. Since every rule captures a specific structural aspect of the training data, we expect any automated post-processing beyond the optimization process to be potentially dangerous. In practice, one could resort to domain experts (e.g., chemists), who are at a better position to judge whether rules are redundant.
>
>
> - **(Q5) Number of Rules Produced for Evaluation Data**
> \
> For Isocyanates, the final, optimized grammar has 34 production rules. There are 78 rules for Acylates and 32 rules for Chain Extenders. For the large polymer dataset, there are in total 341 production rules. The number of production rules is positively correlated with the size of the training set. Despite the fact that a small number of production rules can generate a large number of samples, the constructed grammar becomes more complicated to cover the underlying diversity of the training samples when the training set is larger, resulting in more production rules.
>
>
> - **Minor Issues / Clarifications**
>   - **(Q2) Statement about Performance on More Training Data** \
> Originally, we had additional results about how the model performed with increasing amounts of data, but we removed them because they are not too different from the 0.15% dataset we reported in the paper. However, based on the discussion with reviewer 4wRv, we have added them back to the paper.
>   - **(Q3) Why are some results in the tables underlined?** \
> The underlines highlight second-best results. They are now applied consistently throughout the paper and explained in the captions.

---

> > ### Comment · Reviewer_Qrrj · 2021-11-20
> > **Thanks for the response**
> >
> > Thanks for your effort. You clarified all my doubts, I am now convinced this paper deserves to be accepted and I'm raising my score to 8. Good job!

---

### Official Review · Reviewer_oHTg · 2021-11-03

**Correctness:** 4
**Technical Novelty And Significance:** 3
**Empirical Novelty And Significance:** 4
**Recommendation:** 8
**Confidence:** 4

**Main Review:**

This paper proposes a generic solution for learning to generate graphs in an extreme few-shot manner where only dozens of examples are provided, which is pretty common in the polymer/monomer domain. The method is general and can have a large impact on the graph learning community. Although technically the proposed method is similar to some previous work like MHG[1], it cleverly preserves the class-specific properties by operating on subgraphs instead of individual atoms. Also, the idea of learning a grammar search algorithm using RL instead of directly learning the grammar is novel and interesting in the domain. The proposed method is well-supported by experiments on datasets of different scales.

The paper is well-structured overall but there is still room for improvement. The notation in section 4 is a bit unclear. In section 4.1 the subscripts are usually used for denoting the number of iteration, and the superscripts are used for different connected components. While in section 4.2 right before Equation (3) we see there is $e_t^{(j)}$, where the meaning of superscript $(j)$ is unclear. And $t$ here is used both for denoting the current iteration and the total number of iterations. Also if I understand correctly, $X$ is a binary matrix, and instead of $p(X)=\prod_t\prod_j \phi(e_t^j;\theta)$, it should be $p(X)=\prod_t\prod_j \phi(e_t^j;\theta)^{X_{tj}} (1-\phi(e_t^j;\theta))^{1-X_{tj}}$?

For the experiment section, both Appendix A and Table (3) contain critical information for understanding the experiment setting and results, and so they should be put earlier in the main text.

Question:

1. The paper proposes a clean algorithm for learning grammar. But in practice, there must be a lot of hyperparameters to tune, such as the number of production rules, the maximum/minimum length of a single production rule, etc.? How do you decide these parameters? Do you do any pruning for the production rules?

2. The main text does not mention how to generate molecules using the learned grammar rules. Do you sample the production rules uniformly randomly during testing? If yes, do you plan to learn a policy for generating molecules? How to jointly learn the generation policy together with the rule search policy?

3. The space of grammar rules is combinatorial as mentioned in section 4.2. How does RL circumvent that and learn a reasonable policy?

4. Why do you only use diversity and RS as the optimization objectives, as mentioned in Appendix A?

[1] Hiroshi Kajino. Molecular hypergraph grammar with its application to molecular optimization. In International Conference on Machine Learning, pp. 3183–3191. PMLR, 2019.

**Summary Of The Paper:**

This paper proposes a sample efficient hypergraph grammar learning algorithm for generating molecules. The bottom-up search algorithm conducts iterative contraction according to a learned policy, where in each iteration, several hyperedges are sampled, contracted into a single node, and written into a production rule as part of the final grammar. The search policy is being trained using RL where the rewards are given by evaluating a set of molecules generated by the policy. The proposed method is evaluated on a small and a large monomers dataset and outperforms existing molecule generation algorithms in terms of the synthesizability and the membership rate.

**Summary Of The Review:**

Overall, this paper makes decent contributions in both technical and empirical aspects, and therefore I recommend accepting the paper provided that the authors correct the writing issues.

---

> ### Author Response · Authors · 2021-11-19
> **We thank the reviewer for their helpful comments! Please see our rebuttal below.**
>
>
> - **Formatting**
> \
> We fixed the notation, but after some thoughts, we did not move Appendix A and Table 3 into the main paper. For one reason, this requires substantial space that goes over page limit. More importantly, our approach targets scenarios with small datasets (see also the item "Model Works Only on Small Datasets" in the reply to reviewer LtXp); thus, we only conducted these experiments for comparison. We could shorten the related work but would need to cut additional parts. We wanted to provide this intuition first, however, we are open to suggestions how to fit the contents to page limit.
>
>
> - **(Q1) Hyperparameters**
> \
> Our algorithm actually has no such hyperparameters as you mention (the learned grammar is "... *completely* determined by the sampled order of hyperedges"). We have clarified this in the updated paper. In fact, the algorithm chooses the set of production rules that performs best, without any constraint on the number or the length of the rules.
>
>
> - **(Q1) Pruning**
> \
> We do not do any pruning but keep all (unique) production rules that are learned. In this way, the constructed grammar contains all structural information of the input samples.
>
>
> - **(Q2) Details about Molecule Generation using the Learned Grammar**
> \
> Initially, we tried uniform random sampling of the rules. However, the possibility of generating arbitrarily large molecules (i.e., by choosing production rules with a non-terminal symbol on the right-hand side more likely) sometimes resulted in a never-ending generation process, a problem in practice.
> \
> \
> For that reason, we condition the generation strategy based on (non)terminal symbols in the rules: during generation, we exponentially increase the probability of the production rules without non-terminals symbol on the right-hand side, based on the iteration number. Formally, the probability to select a certain production rule $r$ at iteration $t$ is $p(r) = Z^{-1}exp(\alpha t x_r)$, where $x_r$ is a binary value indicating whether rule $r$ contains only terminal symbols on the right-hand side, and $Z$ is a normalization factor. We used $\alpha=0.5$ in our experiments, since it turned out to reduce the generation time sufficiently while maintaining satisfactory diversity. This part was missing in the submission and we have added it.
> \
> \
> Since we focus on designing a model that can comprehensively represent the molecular design space in this paper, we use this simple generation procedure. It could be replaced by a more advanced one using (reinforcement) learning to select molecules from the design space (e.g., molecules with specific properties). Ideally, as the reviewer suggests, these two components can be jointly learned. This is our immediate future work.
>
>
> - **(Q3) Clarification about Algorithm**
> \
> It is true that finding an optimal grammar amounts to a discrete optimization problem, where the space of grammar rules is combinatorial. MC sampling coupled with REINFORCE algorithm falls in the category of stochastic methods to solve discrete optimization, and it is known to find a global optimum efficiently [2]. In our algorithm, REINFORCE is only used for gradient computation, since the grammar construction process is non-differentiable. In other words, we do not formulate the construction as a Markov decision process and do not use RL.
> \
> [2] Yan, Di, and H. Mukai. "Stochastic discrete optimization." SIAM Journal on control and optimization 30.3 (1992): 594-612.
>
>
> - **(Q4) Choice of Optimization Objectives**
> \
> Generally, our method can support any combination and any number of optimization objectives. In our experiments, we chose diversity and RS for two reasons.
> (I) On the one hand, we consider them to be two of the most important metrics in practice: diversity reflects the comprehensiveness of the generative model, while RS represents the quality of generated molecules. Observe that they nicely complement each other (as can be seen in Fig. 4) and, with both being the objectives, the grammar learning automatically takes this trade-off into account. Lastly, the experimental results illustrate that optimizing only for diversity and RS yields reasonable performance on other metrics as well.
> (II) On the other hand, we want to emphasize the fact that our approach is very different from existing works, which perform distribution fitting based on distribution statistics, such as logP and SA. We are able to consider qualitative metrics which are critical for practical chemical engineering (in our case, judging whether a generative model can indeed discover molecular structures that are both novel and synthesizable).

---

> > ### Comment · Reviewer_oHTg · 2021-11-24
> > **Thanks for the response**
> >
> > Thanks for the detailed response that resolves my concerns! I would maintain the current score and recommend acceptance of the paper.

---

### Official Review · Reviewer_4wRv · 2021-11-05

**Correctness:** 3
**Technical Novelty And Significance:** 4
**Empirical Novelty And Significance:** 3
**Recommendation:** 8
**Confidence:** 3

**Main Review:**

The proposed method is novel and opens potential for new line of research in graph grammar. I personally find the submission to be clear and well-written. Overall, I am positive and would like to give a "accept".

However, I have few more comments that I hope to help improve the paper. Several of them are around the modeling choices and some of them are to clarify my understanding of the practical usefulness of the proposed method.

1. The proposed method uses pretrained graph neural network to generate the sampling weights and claim that this can be replaced by any plug-and-play feature extractor. Can the authors provide data to support this claim? If this is actually true, I would suggest to use the simplest feature extractor to keep the proposed method clean.
2. Is the REINFORCE algorithm stable across random seeds? My hunch is that they are not. Can the authors provide error bars in for their data? Can the authors provide convergence curves to demonstrate the learning?
3. While the proposed method is efficient, I have the impression that it does not scale up well to more data (the authors have to subsample the training set for the experiments in Table 3). My first request is for the authors to report the actual computation time for the proposed method on the experiments they have done. Second, aside from the computational challenge, can the proposed algorithm benefit from more data? If so, can we estimate how much we can gain by including the whole training set of the large polymer dataset?

**Summary Of The Paper:**

This paper introduces a new grammar-based generative model for the molecule generation task. The graph grammar is defined as a set of production rules that operate on module graphs (i.e. the molecule graphs are not linearized as done in some previous work). The graph grammar is learned by iteratively contracting hyperedges (i.e.  edges that can connect multiple nodes, defined by simple chemistry-inspired rules) into non-terminal nodes and the submission proposes to learn how to sample the hyperedges by a REINFORCE algorithm that optimizes for several molecule generation metrics.

The evaluation is done both on small and large molecule generation datasets. Notably, the proposed method achieves strong performance while being very data-efficient.

**Summary Of The Review:**

I find the proposed method to be novel and effective. the paper is clear and well-written. I have a few comments around modeling designs but I think the comments are addressable.

---

> ### Author Response · Authors · 2021-11-19
> **We thank the reviewer for their detailed comments! Please see our rebuttal below.**
>
> - **About Model Components**
>   - **(Q1) Feature Extractor**
> \
> To demonstrate the plug-and-play capability of our system, we ran additional experiments with the most simple embeddings from deepchem [1] (through concatenating atom and bond features). We obtain comparable but slightly worse results (see Appendix D, Figure 6).
> \
> [1] https://deepchem.readthedocs.io/en/latest/api_reference/featurizers.html
>
>   - **(Q2) Stability of REINFORCE** \
> Initial experimental results indicated that the algorithm converged to a similar value across different random seeds, although the trajectories varied to some degree. We have added figures to demonstrate these results in Appendix E.
>
>
> - **(Q3) Scalability** \
> For general considerations regarding the data size, we stress that, unlike what the wealth of DL-based models suggest, the small data scenario is much more common in practice. See the in-depth discussion under "Model Works Only on Small Datasets" in the reply to reviewer LtXp. Although we did not optimize our implementation, which could be substantially accelerated, we note that for the small data scenario, the current runtimes are quite manageable: for the 117-sample training set, our model takes 1 hour per training epoch and we train it for 20 epochs (the 200 in the appendix was an unfortunate typo).
>
>
> - **(Q3) Benefits of More Data**
> \
> We conducted such experiments on 0.3% of the large dataset and obtained only slight performance gains (see the updated Table 3). This can be explained by the nature of our approach. Unlike regular deep learning, every grammar (i.e., even the ones constructed based on very small datasets) fully captures the training data. As a consequence, larger datasets and the resulting grammars are expected to improve only distribution statistics and diversity but not the generally achieved quality.
>  \
>  \
> We can indeed derive an estimate about the obtained gain when training on the entire large polymer dataset, based on the considerations in Jin et al. (2020). The latter work mined a rather comprehensive vocabulary of 436 substructures, which is covered by only (randomly selected) 436 molecules. The 0.15% dataset we consider contains 117 and the above-mentioned 0.30% dataset contains 239 such molecules. Consequently, we expect the distribution statistics to increase slightly when considering all 436 molecules that cover these motifs, but we do not expect major gains if going beyond (that is, including the remaining ~80k training samples for training).

---

### Author Response · Authors · 2021-11-19
**General Response**

We thank all reviewers for the detailed and encouraging reviews! We are grateful that the reviewers recognized the novelty of the topic, the clarity of writing, the fact that our approach addresses serious limitations of existing works, the impressive performance on small datasets, and its potential impact.
\
\
We have addressed specific comments and suggestions in the individual responses and also updated the paper accordingly. In summary, we provided additional experimental results justifying our algorithmic components to reviewer 4wRv, and gave more details and backgrounds about the algorithm to reviewers oHTg and Qrrj. In the reply to reviewer LtXp, we elaborate on the scalability aspect, which was mentioned in several reviews.
\
\
We hope that we have cleared all your concerns and we will be happy to provide further information if needed.

---

### Public Comment · ~Can_Chen3 · 2022-03-06
**A question related to the design**

Hello authors,

Thanks for presenting this interesting work! The graph grammar learning idea is truly fascinating!

I have a naive question. How do you conduct the evaluation?

It seems you can evaluate all metrics easily. If these metrics are very easy to evaluate, why can not you just evaluate more data to augment the limited training dataset?  In most cases, I suppose some properties of proteins can not be evaluated without wet experiments.

I am not familiar with the problem background. Looking forward to your reply!

Best,
Can

---

> ### Public Comment · ~Veronika_Thost1 · 2022-06-21
> **Thank you for your interest in our work!**
>
> Sorry for the late reply, we totally missed this!
>
> The evaluation is done on a set of well-established metrics, typical for evaluating the quality of machine-synthesized molecules. Wet lab evaluation will be attempted in future. Our learned graph grammar provides a means to generate more data for evaluation (as you suggest), and optimization based on the latter.

---

### Decision · Program_Chairs · 2022-01-20

**Decision:**

Accept (Oral)

**Comment:**

This paper presents an approach to learn graph grammars for molecule generation in a very data-efficient way.  The approach combines bottom-up grammar construction by contracting graph substructures and evaluation-driven learning of the parameters in grammar construction in order to optimize metrics of interest.  This paper is well written and the graph grammar learning approach is novel and can potentially have impact beyond just generating molecules.  All reviewers unanimously recommended acceptance of this paper.

A few things emerged in the reviews and discussions with authors, regarding in particular the computational cost and scalability of the approach and the actual molecule sampling process after learning.  I hope the authors can clarify these in the paper, and as the authors said in the discussion that exploring a more expressive model that uses learned probabilities on the production rules can make the model more powerful, which is a promising direction for the future.